# Mitochondrial Genomes of the Blood Flukes *Cardicola forsteri* and *Cardicola orientalis* (Trematoda: Aporocotylidae)

**DOI:** 10.3390/pathogens14070680

**Published:** 2025-07-10

**Authors:** Jemma Hudson, Sunita B. Sumanam, Bronwyn E. Campbell, Lachlan Coff, Barbara F. Nowak, Paul A. Ramsland, Neil D. Young, Nathan J. Bott

**Affiliations:** 1Department of Biology, School of Science, RMIT University, Melbourne, VIC 3000, Australia; s3988903@student.rmit.edu.au (J.H.); bronwyn.campbell@rmit.edu.au (B.E.C.); paul.ramsland@rmit.edu.au (P.A.R.); 2Melbourne Veterinary School, Faculty of Science, The University of Melbourne, Parkville, VIC 3010, Australia; sunita.sumanam@unimelb.edu.au; 3Australian Centre for Disease Preparedness, CSIRO, East Geelong, VIC 3219, Australia; lachlan.coff@csiro.au; 4Institute for Marine and Antarctic Studies, University of Tasmania, Locked Bag 1370, Launceston, TAS 7250, Australia; b.nowak@utas.edu.au; 5Department of Immunology, Monash University, Melbourne, VIC 3004, Australia; 6Department of Surgery, Austin Health, The University of Melbourne, Heidelberg, VIC 3084, Australia

**Keywords:** blood fluke, aquaculture, mitochondrial genome, aquatic animal health, nanopore

## Abstract

Parasitic diseases can be a significant constraint on aquaculture industries, which continue to develop in response to the rise in global demand for sustainable protein sources. Blood flukes, *Cardicola forsteri* and *Cardicola orientalis*, are economically significant parasites of Southern bluefin tuna (Australia), Pacific bluefin tuna (Japan), and Atlantic bluefin tuna (Mediterranean) as they are responsible for blood vessel obstruction in the gills leading to branchitis and mortalities when untreated. Here, we have defined the mitochondrial genomes for these species—the first for any aporocotylids. Oxford nanopore long-read sequencing was used to sequence *C. orientalis* from a single individual. The mitochondrial genome of *C. forsteri* was assembled and curated from available sequence data. Both *Cardicola* spp. mitogenomes contained 12 protein coding, 2 ribosomal and 22 tRNA genes, with the gene order matching that of Asian schistosomes. A control region was identified for each species which contained long and short repeats; the region for *C. forsteri* was longest, and the overall pattern differed between the two species. A surprisingly high nucleotide diversity was observed between the two species, generating interest into the mitochondrial genes of related species. This paper provides a useful resource for future genetics-based research of aporocotylids and other flatworm parasites of socioeconomic significance.

## 1. Introduction

Aporocotylids, or fish blood flukes, are a large family of trematodes comprising >130 species from 35 genera which can cause disease in a range of teleost hosts [1]. Among this group, *Cardicola* spp. have a significant economic impact across tuna industries globally, infecting southern bluefin tuna (SBT, *Thunnus maccoyii*) in Australia, Pacific bluefin tuna (PBT, *Thunnus orientalis*) in Japan, and Atlantic bluefin tuna (ABT, *Thunnus thynnus*) in the Mediterranean [2,3]. The first description of *Cardicola* spp. in Australia was by Cribb et al. [4], who described adult *Cardicola forsteri* from the heart of SBT. Following this, *Cardicola orientalis* was described in the gills of SBT by Shirakashi et al. [5] and retrospectively described as infecting SBT since 2008 by Polinski et al. [6], based on qPCR of historical samples. In Japan, PBT are affected by *C. forsteri* and *C. orientalis* as well as *Cardicola opisthorchis* [7,8]. In the Mediterranean Sea, ABT are affected by all three identified species of *Cardicola*, as well as a fourth *Cardicola* sp. that is currently undescribed due to a lack of morphological data from adult specimens [9]. The pathology of *Cardicola* spp. in bluefin tuna is primarily associated with the presence of eggs within the gills resulting in inflammation of the afferent filaments of the gills [10]. In severe infection this can result in the mortality of affected fish. *Cardicola orientalis* has been thought to cause more severe pathology due to the presence of both adults and eggs in the gills [11]. As pathology can vary among species, it is important to identify which species is present. The morphology of adult *Cardicola* is generally conserved, therefore molecular markers are often used to differentiate between species [12].

The availability of genomic resources is limited for aporocotylids, with a nuclear genome presently available for only one species, *C. forsteri*, and no mitogenome information for any species [13]. The genetic characterisation of *Cardicola* species is currently limited to specific loci used primarily for species identification and diagnostics. The mitochondrial cytochrome oxidase 1 (*cox*1) gene, as well as the mitochondrial ribosomal 12S, 16S, and nuclear 18S, and 28S rDNA are commonly used in trematodes for DNA barcoding and phylogenetic analysis [14,15,16]. In *Cardicola* spp., three ribosomal units (18S, 28S, and ITS-2) and *cox*1 have been utilised for the identification of species globally [5,6,9,17,18]. In particular, the nuclear 28S and ITS-2 rDNA and the mitochondrial *cox*1 have been useful in establishing the current phylogeny of the genus [12].

Mitochondrial genomes are useful for the identification and differentiation of species. Their lack of recombination or genome duplication makes them beneficial for assessing the taxonomy and/or population genetics of parasites [19]. The non-coding control region of parasite mitochondria has been useful in resolving the phylogenetic history of closely related species due to its fast evolutionary rate and highly polymorphic nature [20]. The resolution of this control region has been one of the major methodological challenges in the assembly of trematode mitogenomes, due to the presence of highly repetitive elements and as such, these non-coding regions had been reported as short for many years. The development of longer read sequencing technologies has allowed for the correct annotation of these regions, through the sequencing of reads that can span the full control region without the need for read assembly [21,22,23]. This is yet to be explored in aporocotylids due to the lack of available mitochondrial genome data.

This study aimed to assemble and annotate the mitochondrial genomes for *C. forsteri* and *C. orientalis*, to create a genomic resource for this family, and gain further insight into the phylogenetic relationships of *Cardicola* with other trematodes.

## 2. Materials and Methods

### 2.1. Sample Collection

Whole gill samples were taken from 132 Southern bluefin tuna (SBT), harvested from ranching pontoons in Port Lincoln, South Australia. Sampling was conducted under animal ethics approval (RMIT University Animal Ethics Committee #22802) from harvest fish, euthanised by commercial SBT companies using industry best-practice techniques. The first left gill arch was removed and placed in a sealed bag and stored on ice for 2–4 h. Then, each gill was dissected and flushed with freshwater to dislodge adult flukes using previously described methods [5]. A single *C. orientalis* adult was collected, stored in RNA*later*^®^ (Thermo Fisher Scientific, Scoresby, VIC, Australia) for 24 h at 4 °C and then maintained at −80 °C until further processing.

### 2.2. DNA Extraction (C. orientalis)

For Nanopore long-read sequencing, high molecular weight (HMW) gDNA was extracted from one adult *C. orientalis*, using a previously described schistosome DNA extraction method [24]. In brief, the fluke was rinsed with freshwater to remove excess RNAlater, then homogenised, the protein precipitated and then the DNA precipitated. The extracted DNA (23 ng/µL) was quantified on Qubit 4 Fluorometer (Thermo Fisher Scientific, Waltham, MA, USA). The species was confirmed as *C. orientalis* using ITS-2 rDNA quantitative PCR and the SensiFAST Probe No-ROX Kit, as described previously [25]. Quality and integrity were assessed using a TapeStation system (Agilent 4200) with Genomic DNA ScreenTape (Thermo Fisher Scientific).

### 2.3. DNA Amplification and Size Selection (C. orientalis)

For genomic DNA, 5 µL of extracted HMW DNA was amplified using the Repli-g midi Kit (Qiagen, Hilden, Germany), as per guidelines (SQK-LSK114, Oxford Nanopore Technologies, Oxford, UK), to produce approximately 9.6 µg of DNA. Amplified DNA was sheared 5 times using a 26 G needle and a 1 mL syringe for preparing long DNA fragments for nanopore library preparation. The integrity of the amplified genomic DNA was verified using an Agilent 4200 TapeStation system (Thermo Fisher, Waltham, MA, USA) and Genomic DNA ScreenTape (Thermo Fisher, Waltham, MA, USA).

### 2.4. Library Preparation and Sequencing

For library preparation, 4.2 μg of the sheared amplified gDNA was subjected to T7 endonuclease treatment according to guidelines (SQK-LSK114, Oxford Nanopore Technologies). Final DNA concentration was measured using a Qubit 4 Fluorometer (Thermo Fisher Scientific, Waltham, MA, USA). Approximately 1 μg of DNA was used to construct a library, according to the protocol (SQK-LSK114, Oxford Nanopore Technologies). Approximately 150 ng of the constructed library was then sequenced for 24 h on a 10.4 flow cell and using a MinION Mk1B (Oxford Nanopore Technologies). The flow cell was washed using Flow cell wash kit (EXP-WSH004, Oxford Nanopore Technologies) and an additional 150 ng of library was loaded and sequenced for ~117 h, until the flow cell pores were depleted. Base-calling of the raw Fast5 file was performed using the program Guppy v.6.4.6 (Oxford Nanopore Technologies) (model: dna_r10.4.1_e8.2_400bps_sup@v5.2.0) and saved in FASTQ format.

Raw *C. forsteri* short-read (Illumina) and long-read (Oxford Nanopore, Oxford UK) data were publicly available (NCBI: PRJNA810749); as per Coff et al. [13] this data were obtained from adult *C. forsteri* sampled from the hearts of SBT collected in Port Lincoln, South Australia. The *C. forsteri* FAST5 raw data were base-called using Guppy v1.2.0 (Oxford Nanopore Technologies) and stored in FASTQ format. PycoQC v2.5.2 reports were created for both *C. forsteri* and *C. orientalis*.

### 2.5. Assembly and Annotation of Mitochondrial Genomes

The mitogenome of *C. orientalis* was assembled using the MinION data using CANU v.2.3 [26]. First, reads were mapped to publicly available Schistosomatidae reference mitochondrial genomes using Minimap2 v2.26 [27]. Reads with homology to a reference mt genome were used for assembly. Mapped reads from the MinION read data were used to polish the assembled *C. orientalis* reference genome and estimate coverage. Once assembled, the package Mitos 2.0.2 was used for annotation, then the mitogenome was rotated to position *cox*1 as the first gene.

Long reads for *C. forsteri* were mapped to the *C. orientalis* mt genome using Minimap2 v2.26 [27]. Reads mapped to *C. orientalis* were retained and used for mt genome assembly. The package FLYE v2.9.2 [28] was used to assemble the mt genome and 5 polishing runs were performed to correct any errors. Once assembled, the mitogenome was rotated to start at the gene *cox*1. The software fastp v0.23.4 [29] was used to filter the Illumina reads, and quality control was performed on the filtered reads. The filtered Illumina reads were used to polish the assembled mt genome using the package HyPo v1.0.3 [30], and Minimap2 to map the short and long reads to contigs.

### 2.6. Control Region Length

The lengths of the control regions were confirmed with a consensus of reads mapping the full length of the region for each species, identified using Minimap2. Repeat regions were predicted in control regions using a repeat-match tool in mummer v.3.23 [31] and manually curated. The annotation of the control region with curated repeat library was performed using RepeatMasker v4.1.5. Repeat regions that occurred >5 times were retained for further analysis.

### 2.7. Nucleotide Diversity

A sliding window analysis was performed of the alignment of the coding regions for the two species, using a window size of 300 and a step size of 10. Analysis was performed in R (RStudio-2024_04) [32]. A distance matrix was made using Ape v.5.8.1 [33] and sliding window analysis performed with the package Spider v.1.5 [34].

### 2.8. Phylogenetics

25 *cox*1 sequences of highly related species were identified though a NCBI blast search (Appendix A) and aligned to the *cox*1 sequences of *C. forsteri* and *C. orientalis* using clustal omega [35] in Geneious prime 2023.2.1. The partial *cox*1 region was chosen as full mitochondrial genomes were not available for most species. The aligned sequences were trimmed at regions with gaps due to missing data. ModelFinder [36] was used to identify an appropriate nucleotide substitution model. A maximum likelihood phylogenetic tree was constructed using the Kimura model K81 [37] in IQ-TREE (v2.3.5) [38] and using ultrafast bootstrap [39] with 1000 replicates. The final tree was presented as a 50% majority-rule tree.

## 3. Results

The final sequencing run for *C. orientalis* yielded a total of 5,750,829 reads, with a N50 of 4733 bp. The circular mitochondrial genomes for *C. forsteri* and *C. orientalis* were 34,397 bp and 28,297 bp in size, respectively, with coding regions of 13,890 bp and 13,900 bp, and a single non-coding region measuring 20,510 bp in *C. forsteri* and 14,397 bp in *C. orientalis* (Figure 1). A total of 36 genes were found for each species (Table 1 and Appendix A), including 12 protein coding genes, 22 transfer RNA and 2 ribosomal RNA. The gene order was the same for both species, with all genes transcribed in the same direction. All genes used ATG or GTG start codons, and most genes used TAA or TAG stop codons, as common for trematodes [40,41]. For *cox*1 and *nad*4L in both species, and *cox*2, *nad*5, and *atp*6 in *C. orientalis*, abbreviated stop codons are present, with a single T nucleotide following the last complete codon of each gene. The nucleotide length of each coding gene was similar between the two species, with none having a difference in length greater than 42 bp. The largest difference was seen in *nad*2, which was longer in *C. orientalis* compared to *C. forsteri*. Three genes were identical in length across the two species: *cox*3, *nad*4, and *nad*1. The nucleotide content of both species was highest for T, with C as the least favoured nucleotide. However, *C. forsteri* had a higher relative G% content, while *C. orientalis* has a higher C% content (Table 2). The overall nucleotide diversity in the coding region was high between the two species (Figure 2). The genes with the highest diversity were *cox*2 (π = 0.38), *nad*5 (π = 0.43), *nad*2 (π = 0.36), *atp*6 (π = 0.50), and *nad*4 (π = 0.40). The lowest nucleotide diversity was seen in *cox*1 (π = 0.25), *nad*1 (π = 0.28), and *nad*3 (π = 0.29).

A long non-coding region (NCR) was identified for both species, with the region being longer in *C. forsteri* at 20,510 bp compared to 14,397 bp in *C. orientalis* (Figure 1). The length of this NCR was confirmed for *C. orientalis* from the average of 3963 reads which spanned across the complete NCR, including protein-coding gene elements on either side. Five reads were mapped across the complete NCR for *C. forsteri*, including protein-coding gene elements on either side, and the longest was taken as the inferred length. For each species, five repeats were identified; however, the length of each repeat and the pattern differed between the two species. For example, the longest repeat found in each species were similar lengths at CfR3 (665 bp) and CoR2 (666 bp) in *C. forsteri* and *C. orientalis*, respectively, however the frequency of each was substantially different (Table 3). Similarly, the shortest repeats for each species are of similar lengths: CfR4 (69 bp) and CoR3 (70 bp), also with different frequencies (Table 3). The *C. forsteri* repeat region follows a pattern of short-short-long with the CfR1, CfR2, and CfR3 repeats that continues along most of the control region, repeating 16 times (Figure 1). *C. orientalis* has a similar short-short-long repeat pattern, however with only two different repeats: two CoR1 units, followed by one CoR2. This section is also a shorter part of the *C. orientalis* control region, repeating 7 times, followed by a longer pattern of short repeats. The repeats form hairpin loop structures, with high A and T nucleotide content (Figure 1). In *C. forsteri* the repeats CfR2, CfR4, and CfR5 form single stem-loop structures, while CfR1 and CfR3 form more complex repetitive hairpin structures (Appendix A). In *C. orientalis* only one repeat, CfR4, forms a single stem-loop structure (Appendix A).

The *cox*1 phylogenetic tree was able to verify the position of the two *Cardicola* species in relation to previously characterised groups. The tree revealed strong nodal support for three groups, with *C. forsteri* and *C. orientalis* grouping together with a more recent relationship of *C. forsteri* to *C. opisthorchis* (Figure 3). There was also support for a close ancestral relationship between the *Cardicola* and *Paradeontacylix*. Poor nodal support was observed for a relationship between these two groups with schistosomatids.

## 4. Discussion

This paper presents the first mitochondrial genomes for any members of the Aporocotylidae and shows the presence of long tandem-repeat regions in both. We demonstrated that the full mitochondrial genome could be sequenced from a single specimen using long-read sequencing, which is highly beneficial as it reduces issues of heterozygosity and shows potential for genomics work in species where little genetic material is available. The mitochondrial genomes of *C. forsteri* and *C. orientalis* are consistent with other trematodes, containing all 36 genes common to trematodes, and lacking *atp*8. The gene order is the same as Asian schistosomes such as *Schistosoma japonicum* and *S. mekongi* with *nad*1 preceding *nad*3, and *atp*6 and *nad*2 between *nad*4 and *nad*1 [42,43]. The length of the two mitogenomes is longer than what is reported for most trematodes; however, this is mostly due to the extensive non-coding region identified, which is significantly larger than most previously reported NCR [44]. The coding region, however, falls within the expected range, slightly longer than several species including *Fasciola hepatica*, *Hypoderaeum conoideum*, *Echinostoma caprioni*, *Paragonimus westermani*, and *Prosthogonimus cuneatus*, but shorter than most schistosomes such as *Schistosoma spindale*, *S. mansoni*, *S. japonicum*, and *S. mekongi* [44,45,46].

The protein coding genes are similar across the two species, with comparable lengths and gene boundaries. A small overlap of 2 bp is present between the *nad*4L and *nad*4 genes for both *C. orientalis* and *C. forsteri*. This overlap is common across trematodes, though the overlap here is much less than what has previously been observed [47,48]. Two abbreviated stop codons are present in *C. forsteri* in *cox*1 and *nad*4L, and five in *C. orientalis* in *cox*1, *nad*4L, *cox*2, *nad*5, and *atp*6: each with an overhanging T nucleotide. Polyadenylation—where a triplet A tail is added during transcription, allowing incomplete T or TA nucleotides to become stop codons—is relatively common in metazoan mitogenomes, and has been observed in several trematodes [49,50,51,52]. Only one gene was significantly different in length between the two species—*nad*2, which was 42 bp longer in *C. orientalis* compared to *C. forsteri*. A gap in the sequence between the end of *atp*6 and the start of *nad*2 was observed in *C. forsteri*, whereas none was seen in *C. orientalis*. It is possible that a single nucleotide sequence error in the *C. forsteri* mitochondrial genome resulted in a missing start codon upstream of where the current *nad*2 annotation is located. In this case, a GCG is present immediately after the *atp*6 stop codon; if this were a GTG it could act as the start codon for *nad*2, resulting in genes of similar lengths for both species. The overall lengths of the protein coding genes for both species fall within the range seen for trematodes [43].

Despite sharing multiple host species, *C. forsteri* and *C. orientalis* are genetically distinct, with low sequence identity. The nucleotide content of the protein coding region is low in C and high in T for both species, which is typical for trematodes; however, the A and G content are higher and lower, respectively, than what is seen for most species [53,54,55,56,57]. The high sequence variation may indicate that many of the genes could be useful in species identification between *C. forsteri* and *C. orientalis*; however, the design of universal primers may be difficult with these genes. The relatively lower nucleotide diversity of *cox*1, *nad*1, and *nad*3, may make them the most useful for the design of universal primers.

Compared to the nucleotide sequence identity of other species that parasitise similar hosts, the level of variation in *C. forsteri* and *C. orientalis* is surprisingly high. For example, *Azygia robusta* and *A. hwangtsiyui* (Plagiorchiida: Azygidae)—trematode parasites of elasmobranchs and freshwater teleosts—have 26.95% sequence variation in their mitogenomes [53]. Even *Schistosoma bovis* and *Schistosoma haematobium*, which infect different organs in different intermediate hosts, share 97% sequence identity suggesting a likelihood for interspecies hybridisation [58]. For the two *Cardicola* species reported here, the high nucleotide variation could indicate clear species boundaries. Future research to sequence the mitochondrial genomes of other *Cardicola* species would be useful, to explore the evolutionary relationships within this genus. Particularly, having the mitochondrial genome for *C. opisthorchis* and the undescribed Mediterranean *Cardicola* sp. would be interesting to see if there is similar nucleotide diversity across all four species infecting bluefin tuna worldwide.

The non-coding region was significantly different between the two species, both in length and pattern of repeats. This region is usually where variation is found in trematode mitogenomes, with differences on both inter- and intra-species levels [44,59]. For example, specimens of *Paragonimus ohirai* and *Echinochasmus japonicus* have shown different lengths of NCR in individuals from different locations, due to a differing number of repeats [55,60]. Polymorphism was also observed in the *P. westermani* repeat region that suggested variation within individual worms [61]. The identification of long non-coding regions is still only recent, with the development of long read sequencing allowing these regions to be elucidated without the need for assembly, which was difficult due to the repetitive nature of the NCR. The length of the *C. orientalis* NCR is within the range of long NCRs that have been identified so far: non-coding regions of 4–5 kb have been identified in *S. bovis* and *Clonorchis sinensis*, 6.9 kb in *P. westermani*, and 18.5 kb in *S. haematobium* [23,58,61]. The NCR of *C. forsteri* is now the longest NCR to be identified for a trematode to date at >20 kb. Despite the variation between the two species, the identification of these NCR is highly confident as they were identified without the need for assembly. For both species, the whole NCR was confirmed through a consensus of reads mapping along the entire region. In *C. orientalis* several thousand reads were identified that covered the full length of the region; however, even in *C. forsteri* in which no amplification was performed, 5 reads covered the full length of the non-coding region. The identification of these long NCRs in two *Cardicola* species further highlights the benefit of high quality long-read sequencing in assembling genomes and adds to the idea that the length of non-coding regions in mitochondrial genomes is longer than previously thought.

The *cox*1 phylogenetic tree presented here supports the position of the two *Cardicola* species in relation to previously characterised groups, in particular showing a close relationship with *Paradeontacylix*, which had been previously suggested with partial lsrDNA data [62]. Only partial *cox*1 sequences were available for many aporocotylids, many of which did not cover the same region, meaning that several Aporocotylidae were left out of the tree presented here. Detailed phylogenetic analysis of the Aporocotylidae would require greater taxonomic representation of species at the complete mitochondrial genome level. Future studies generating wider taxon sampling and sequencing would greatly improve the ability to understand the evolutionary history and phylogenetic relationships of these parasites.

## 5. Conclusions

The mitochondrial genomes presented here add to the database of genomic information for parasitic trematodes and are the first mitogenomes of any aporocotylids. The high variation between the two species despite their common locality and shared host makes the investigation of the other two species of *Cardicola* that infect bluefin tuna globally, *C. opisthorchis* and *Cardicola* sp., crucial. Additionally, an investigation of population differences that may occur between all *Cardicola* spp. globally is now warranted. Overall, our research highlights the need for more genetic data for aporocotylid trematodes.

## Figures and Tables

**Figure 1 pathogens-14-00680-f001:**
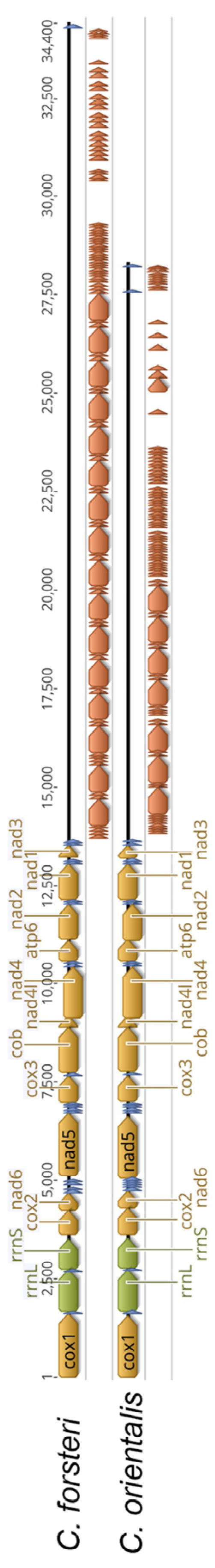
Linear view of the complete mitochondrial genomes of *C. forsteri* and *C. orientalis*. Block arrows indicate genes according to colour: yellow—protein coding, green—rRNA, blue—tRNA, orange—tandem repeats. Whole genome length of *C. forsteri*: 34,398 bp; and *C. orientalis*: 28,297 bp. Tandem repeat region length of *C. forsteri*: 20,510 bp; and *C. orientalis*: 14,397 bp.

**Figure 2 pathogens-14-00680-f002:**
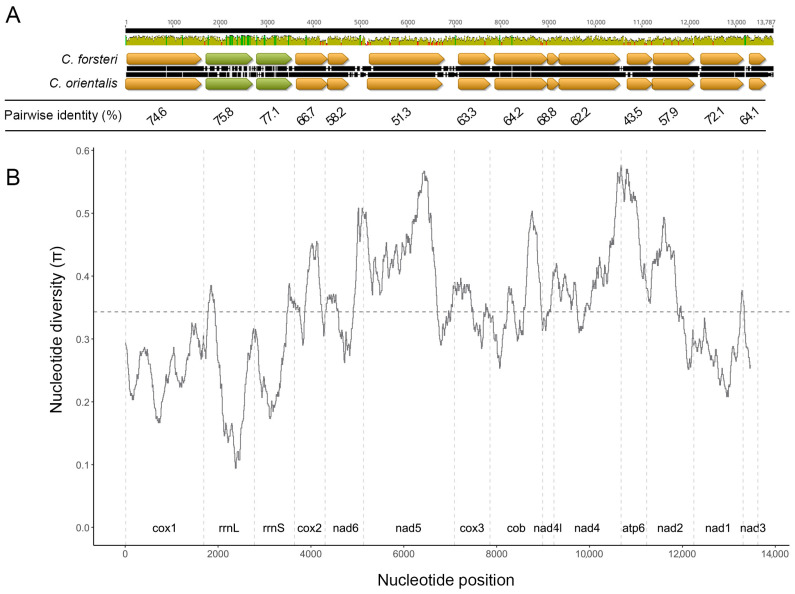
Nucleotide diversity of the coding regions of *Cardicola forsteri* and *Cardicola orientalis.* (**A**). Histogram plot shows pairwise identity of the coding region (green = 100%, yellow = 30–100%, red < 30%), with yellow arrows representing protein coding genes, and green arrows representing rRNA genes. Black bars between the gene sequences represent disagreements between nucleotides. Mean pairwise identity for each gene is presented below this plot. (**B**). Nucleotide diversity across the genome (window size = 300 bp, step size = 10 bp). The horizontal dotted line denotes the mean nucleotide diversity, and vertical dotted lines represent gene boundaries.

**Figure 3 pathogens-14-00680-f003:**
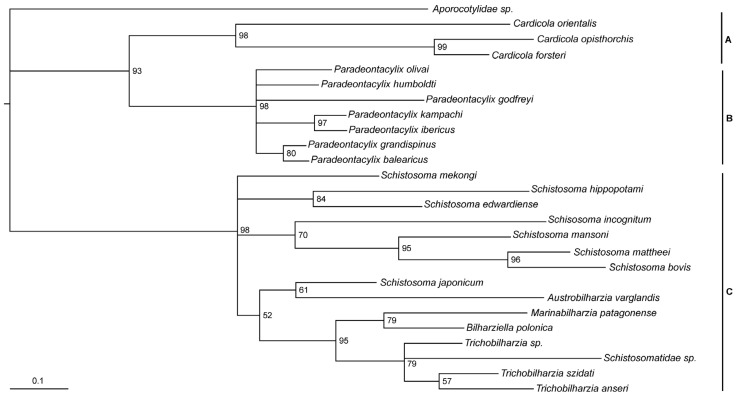
Partial *cox*1 phylogenetic tree of *Cardicola forsteri* and *Cardicola orientalis* with related trematode species. Grouping of closely related species represented by letters on right side of tree: (A) *Cardicola* spp., (B) *Paradeontacylix* spp., (C) Schistosomes, with *Aporocotylidae* sp. resolved to be the outgroup. Bootstrap support for each is not shown. The scale bar represents substitutions per site.

**Table 1 pathogens-14-00680-t001:** Comparison of protein coding gene length and codon usage in the mitochondrial genomes of *Cardicola forsteri* and *Cardicola orientalis*. Abbreviated stop codons are represented by two asterisks (**) following the final nucleotide of the gene.

Gene	Nucleotide Length		Start Codon		Stop Codon	
	*C. forsteri*	*C. orientalis*	*C. forsteri*	*C. orientalis*	*C. forsteri*	*C. orientalis*
*cox*1	1591	1615	GTG	ATG	T **	T **
*cox2*	660	661	GTG	GTG	TAG	T **
*nad*6	438	459	GTG	GTG	TAG	TAG
*nad*5	1581	1579	ATG	GTG	TAG	T **
*cox*3	675	675	ATG	ATG	TAG	TAG
*cob*	1134	1125	GTG	ATG	TAG	TAG
*nad*4L	247	244	ATG	ATG	T **	T **
*nad*4	1293	1293	ATG	ATG	TAA	TAA
*atp*6	525	526	ATG	ATG	TAA	T **
*nad*2	861	903	ATG	ATG	TAG	TAG
*nad*1	903	903	ATG	ATG	TAA	TAA
*nad*3	336	339	GTG	ATG	TAG	TAG

**Table 2 pathogens-14-00680-t002:** Comparison of protein coding gene nucleotide composition (%) in the mitochondrial genomes of *Cardicola forsteri* and *Cardicola orientalis*.

Gene	A		C		G		T	
	*C. forsteri*	*C. orientalis*	*C. forsteri*	*C. orientalis*	*C. forsteri*	*C. orientalis*	*C. forsteri*	*C. orientalis*
*cox*1	20.7	21.6	10.2	15.1	27.0	22.3	42.1	41.0
*cox2*	21.5	22.7	10.2	13.3	31.2	25.3	37.1	38.7
*nad*6	20.3	20.0	7.5	14.2	28.5	22.7	43.6	43.1
*nad*5	18.8	24.1	8.6	13.4	30.6	22.3	42.1	40.2
*cox*3	19.0	21.8	7.3	12.1	28.4	21.9	45.3	44.1
*cob*	19.3	23.6	10.8	15.4	27.7	21.1	42.2	39.9
*nad*4L	22.3	22.5	6.9	10.7	34.0	23.8	36.8	43.0
*nad*4	20.1	21.0	10.9	16.6	28.8	23.2	40.2	39.2
*atp*6	22.3	27.0	12.4	18.8	27.6	15.0	37.7	29.2
*nad*2	18.5	22.6	9.8	16.1	28.7	21.3	43.1	40.1
*nad*1	20.6	22.9	9.7	13.0	29.0	22.3	38.8	41.9
*nad*3	20.2	25.4	9.2	15.9	30.4	20.4	40.2	38.3
*Full coding region*	21.8	24.2	10.3	14.5	28.4	22.4	39.5	38.9

**Table 3 pathogens-14-00680-t003:** Tandem repeats in the control regions of *Cardicola forsteri* and *Cardicola orientalis* mitogenomes.

*C. forsteri*			*C. orientalis*		
Repeat Name	Length (bp) ^1^	Frequency	Repeat Name	Length (bp) ^1^	Frequency
CfR1	104	24	CoR1	66	52
CfR2	77	22	CoR2	666	7
CfR3	665	20	CoR3	70	15
CfR4	69	22	CoR4	79	12
CfR5	128	13	CoR5	371	8

^1^ Length based on most common length of repeat.

## Data Availability

Complete mitochondrial genome sequences have been deposited in the GenBank database under the accession nos. PV670778 and PV670779.

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
