# Peer review of "Mitochondrial Genomes of the Blood Flukes Cardicola forsteri and Cardicola orientalis (Trematoda: Aporocotylidae)"

_pathogens, 2025, doi:10.3390/pathogens14070680_

Round 1
Reviewer 1 Report
Comments and Suggestions for Authors
Introduction
L. 40-41. Could you please provide the scientific name of each fish?
L. 46. Mediterranean Sea?
Materials and Methods
Please provide details of the locality where the other specie of Cardicola was sampled?
Results
L. 168-169. How many Non Coding Regions?
How many amino acids in the PCG and which are the most frequent?
L. 182-183. This indicates that cox1, nad1 and nad3 are the most useful genetic markers for genetic lineage determination and species identification?
Discussion
L. 250-251. how many common flatworm mitochondrial genes contains?
Some minor errors must be corrected
L. 241. Cardicola spp. and Paradeontacylix spp., spp. Should not be italic
L. 242. Aporocotylidae sp. spp. Should not be italic
L. 364. Nowak, B. homogenize Nowak, B.F.
L. 390. International Journal for Parasitology. Must be abbreviated
L. 390. The correct spelling is Pérez-Ponce de León, G.
L. 396. The accurate spelling is Muñoz
L. 408. International Journal of Molecular Sciences. Must be abbreviated
L. 418. Nowak, B. homogenize
L. 424. Nature Biotechnology. Must be abbreviated
L. 510. International Journal for Parasitology. Must be abbreviated
Author Response
Reviewer 1:
- 40-41. Could you please provide the scientific name of each fish?
Thanks, this has been added. (Lines 40-41)
- 46. Mediterranean Sea?
Thanks, this has been changed. (Line 47)
Please provide details of the locality where the other species of Cardicola was sampled?
We have updated the methods to include this detail. The text now reads: “Raw C. forsteri short-read (Illumina) and long-read (Oxford Nanopore) data were publicly available (NCBI: PRJNA810749); as per Coff et al. [13], this data was obtained from adult C. forsteri sampled from the hearts of SBT collected in Port Lincoln, South Australia.” (Lines 128-130)
- 168-169. How many Non-Coding Regions?
The mitochondrial genomes presented here each contain one non-coding region. This has been clarified in the text, which now reads: “The circular mitochondrial genomes for C. forsteri and C. orientalis were 34,397 bp and 28,297 bp in size respectively, with coding regions of 13,890 bp and 13,900 bp, and a single non-coding region measuring 20,510 bp in C. forsteri and 14,397 bp in C. orientalis (Figure 1).” (Lines 178-179)
How many amino acids in the PCG and which are the most frequent?
The length of the protein coding region has already been provided for both species (Line 178), as well as the most frequent amino acid (T) (Line 190). For clarity, we have added the nucleotide composition for the full coding region to Table 2 (Line 218).
- 182-183. This indicates that cox1, nad1 and nad3 are the most useful genetic markers for genetic lineage determination and species identification?
Yes, these markers may be the most useful for genetic lineage determination, although the high diversity of all genes may make any of them useful in species identification. Two lines have been added to the discussion to clarify this: “The high sequence variation may indicate that many of the genes could be useful in species identification between C. forsteri and C. orientalis, however the design of universal primers may be difficult with these genes. The relatively lower nucleotide diversity of cox1, nad1 and nad3, may make them the most useful for the design of universal primers.” (Lines 294-297)
- 250-251. how many common flatworm mitochondrial genes contains?
The mitochondrial genomes for C. forsteri and C. orientalis contain all the common flatworm mitochondrial genes (36 total). This sentence has been reworded to clarify: “The mitochondrial genomes of C. forsteri and C. orientalis are consistent with other trematodes, containing all 36 genes common to trematodes, and lacking atp8.” (Lines 262-263)
- 241. Cardicola spp.and Paradeontacylix spp., spp. Should not be italic
- 242. Aporocotylidae sp. spp. Should not be italic
Thanks for pointing this out, this has been fixed. (Line 253, Line 254)
- 364. Nowak, B. homogenize Nowak, B.F.
- 418. Nowak, B. homogenize
Thanks for noticing these errors, they have now been fixed. (Line 382, Line 435)
- 390. International Journal for Parasitology. Must be abbreviated
- 408. International Journal of Molecular Sciences. Must be abbreviated
- 424. Nature Biotechnology. Must be abbreviated
- 510. International Journal for Parasitology. Must be abbreviated
Thanks, these have now been abbreviated (Line 407, Line 425, Line 441, Line 529)
- 390. The correct spelling is Pérez-Ponce de León, G.
Thank you, this has been corrected. (Line 408)
- 396. The accurate spelling is Muñoz
Thanks, this has been corrected (Line 413)
Reviewer 2 Report
Comments and Suggestions for Authors
The assessed work is the first study on the mitochondrial genome of two extremely important fluke species that have a major impact on tuna farming worldwide.
The research was carried out by an experienced scientific team. The publication fills a gap in the genetic database of these important flukes from an economic point of view.
The authors demonstrated extensive knowledge of issues from both the field of molecular biology and prazitology. This is confirmed by a very accurate discussion and the selection of cited source articles. I do not make any comments. My possible suggestions would be only my personal view of issues related to the invasiveness of the analyzed flukes, which will not affect the result and its interpretation. The most important aspect of the article is the genetic differentiation within the genus and also the Digenea Class. I consider the study to be very valuable.
Author Response
Thank you as well to reviewer 2, for your comments on the manuscript, and for giving your time in reviewing our work.
Reviewer 3 Report
Comments and Suggestions for Authors
The article presents the first data on the complete mitochondrial genome not only of the genus Cardicola but also of the family Aporocotylidae, which underscores its undeniable novelty. The importance of this information is also indisputable, as it enables the development of new and the improvement of existing methods for precise PCR diagnostics of practically significant parasites such as trematodes of the genus Cardicola. These parasites are pathogens of commercially valuable fish species, including tunas. All of this makes the article worthy of publication.
I have no major remarks regarding the structure of the article, the completeness of the methods and results presented.
Minor remarks:
- Line 153: It is likely necessary to cite RStudio Team, 2024 as [33] and add to the reference list something like: RStudio Team (2024). RStudio: Integrated Development Environment for R. RStudio, PBC, Boston, MA. URL: http://www.rstudio.com/.
- Line 305: The full species name should be given as Chlonorchis sinensis, as it is misleading to imply that it belongs to the genus Cardicola. Also, check the reference number, as it seems that in your reference list, it is 24, not 22: 24. Kinkar, L.; Young, N.D.; Sohn, W.M.; Stroehlein, A.J.; Korhonen, P.K.; Gasser, R.B. First record of a tandem-repeat region within the mitochondrial genome of Chlonorchis sinensis using a long-read sequencing approach. PLoS Negl Trop Dis 2020, 14, e0008552. doi:10.1371/journal.pntd.0008552.
Given this error, please verify all citation numbers in the text. - Line 305: Starting a sentence with "At >20 kb,…" is not ideal, as it looks like an abbreviation. Consider rewording.
- Lines 326–332: This paragraph reads like a conclusion or summary, partially reiterating the previous discussion, so it would be better to label it as "Conclusions".
Author Response
Reviewer 3:
Line 153: It is likely necessary to cite RStudio Team, 2024 as [33] and add to the reference list something like: RStudio Team (2024). RStudio: Integrated Development Environment for R. RStudio, PBC, Boston, MA. URL: http://www.rstudio.com/.
Thank you, the reference for RStudio has been added. (Lines 449-450)
Line 305: The full species name should be given as Chlonorchis sinensis, as it is misleading to imply that it belongs to the genus Cardicola. Also, check the reference number, as it seems that in your reference list, it is 24, not 22: 24. Kinkar, L.; Young, N.D.; Sohn, W.M.; Stroehlein, A.J.; Korhonen, P.K.; Gasser, R.B. First record of a tandem-repeat region within the mitochondrial genome of Chlonorchis sinensis using a long-read sequencing approach. PLoS Negl Trop Dis 2020, 14, e0008552. doi:10.1371/journal.pntd.0008552.
Given this error, please verify all citation numbers in the text.
Thanks for pointing this out, the full name has now been provided (Line 322), and the references updated throughout.
Line 305: Starting a sentence with "At >20 kb,…" is not ideal, as it looks like an abbreviation. Consider rewording.
We have reworded this sentence to read: “The NCR of C. forsteri is now the longest NCR to be identified for a trematode to date at > 20 kb.” (Lines 322-323)
Lines 326–332: This paragraph reads like a conclusion or summary, partially reiterating the previous discussion, so it would be better to label it as "Conclusions".
Thank you for the suggestion, this paragraph has been labelled as “Conclusions”. (Line 343)
Reviewer 4 Report
Comments and Suggestions for Authors
Dear Editor,
This paper presents a mitochondrial genome analysis for two aporocotylid parasites Cardicola forsteri and C. orientalis, which infect Thunnus taxa. This work addresses an important gap in the genomic data available for these taxa. For C. orientalis , authors used Long-read sequencing approach which is effective for resolving repetitive regions sequencing.
However, I have several important comments and suggestions:
One major concern, the authors state within the abstract line 22-24 that this is the first time sequencing has been performed for Cardicola forster. However, these data were published previously, reference 13, please correct this statement accordingly in the manuscript also in the title.
Line 26: “A long repetitive control region was identified for each species” do you mean satellite DNA? Which length? Consider clarifying the term "long" meaning
Line 47: “as well as a fourth undescribed Cardicola sp. [9].” Can you please develop this information why this species couldn’t be identified, the approach used….
Can you also add two sentences mentioning the methodological challenges specific to trematodes genome assembly.
Line 92: add more details about the DNA extraction method.
Line 100: the Repli-g midi kit typical yield 40 µg per reaction. Can you explain why in your case it produced only 9.6 µg?
Add more details about the utility to use the Repli-g 100 midi Kit (Qiagen, Germany)
Line 108: “200 μL of the sheared amplified gDNA” better use quantities or concentration.
Line 116: how did you choose the sequencing time (117h)
Line 124: why are you repeating the analysis for C. forsteri while you published the work previously?
Line 156: why did you use only a fragment of the COI instead of the whole mitochondrial genome?
Line 166; Indicate the total yield of reads from the sequencing run.
typically the WMG length in trematodes ranges from 14,000 to 16,000 base pairs. can you explain the unusual result? Were assembly and circularization validated? There is no information about the short vs long read results in the case of Cardicola forsteri.
Author Response
One major concern, the authors state within the abstract line 22-24 that this is the first time sequencing has been performed for Cardicola forsteir. However, these data were published previously, reference 13, please correct this statement accordingly in the manuscript also in the title.
Thank you for the suggestion, the relevant sentence in the abstract has been reworded accordingly, and now reads: “The mitochondrial genome of C. forsteri was assembled and curated from available sequence data.” (Lines 23-24)
Line 26: “A long repetitive control region was identified for each species” do you mean satellite DNA? Which length? Consider clarifying the term "long" meaning
This has been clarified to read: “A control region was identified for each species which contained long and short repeats; the region for C. forsteri was longest, and the overall pattern differed between the two species.” (Lines 26-27)
Line 47: “as well as a fourth undescribed Cardicola sp. [9].” Can you please develop this information why this species couldn’t be identified, the approach used….
This sentence has been expanded to read: “In the Mediterranean Sea, ABT are affected by all three identified species of Cardicola, as well as a fourth Cardicola sp. that is currently undescribed due to a lack of morphological data from adult specimens [9].” (Lines 48-49).
Can you also add two sentences mentioning the methodological challenges specific to trematodes genome assembly.
Yes, thank you for the suggestion. The major challenge in trematode mitochondrial genome assembly is the presence of the repeat-rich control region. We have expanded the relevant section of the introduction, which now reads: “The non-coding control region of parasite mitochondria has been useful in resolving the phylogenetic history of closely related species due to its fast evolutionary rate and highly polymorphic nature [21]. The resolution of this control region has been one of the major methodological challenges in the assembly of trematode mitogenomes, due to the presence of highly repetitive elements and as such, these non-coding regions had been reported as short for many years. The development of longer read sequencing technologies has allowed correct annotation of these regions, through the sequencing of reads that can span the full control region without the need for read assembly [22-24].” (Lines 71-77)
Line 92: add more details about the DNA extraction method.
A statement was added to summarise the methods followed: “For Nanopore long-read sequencing, high molecular weight (HMW) gDNA was extracted from one adult C. orientalis, using a previously described schistosome DNA extraction method [25]. In brief, the fluke was rinsed with freshwater to remove excess RNAlater, then homogenized, the protein precipitated and then the DNA precipitated.” (Lines 97-98)
Line 100: the Repli-g midi kit typical yield 40 µg per reaction. Can you explain why in your case it produced only 9.6 µg?
Yes, this is a result of the yield for the Repli-g midi kit being dependent on the the starting template. In our case, the starting template was a very low concentration of DNA from a single worm, and as such a lower yield was anticipated.
Add more details about the utility to use the Repli-g 100 midi Kit (Qiagen, Germany)
This kit was used to amplify the extracted DNA, as per the recommendation of the Oxford Nanopore ligation sequencing gDNA whole genome amplification protocol (SQK-LSK114). The relevant section of the text has been updated to read: “For genomic DNA, 5 µL of extracted HMW DNA was amplified using the Repli-g midi Kit (Qiagen, Germany), as per guidelines (SQK-LSK114, Oxford Nanopore Technologies), to produce approximately 9.6 µg of DNA.” (Lines 107-108)
Line 108: “200 μL of the sheared amplified gDNA” better use quantities or concentration.
Thank you for the suggestion, this has been changed in the text: “For library preparation, 4.2 μg of the sheared amplified gDNA was subjected to T7 endonuclease treatment according to guidelines (SQK-LSK114, Oxford Nanopore Technologies).” (Line 115)
Line 116: how did you choose the sequencing time (117h)
The sequencing was run until the pores from the flow cell had been depleted. This has been updated in the text as follows: “The flow cell was washed using Flow cell wash kit (EXP-WSH004, Oxford Nanopore Technologies) and an additional 150 ng of library was loaded and sequenced for ~ 117 h, until the flow cell pores were depleted.” (Line 123)
Line 124: why are you repeating the analysis for C. forsteri while you published the work previously?
Although we are using the raw sequencing data generated previously (as per Coff et al. 2022), this is the first time the mitochondrial genome for C. forsteri is being assembled and annotated.
This has now been clarified in the text, as per previous comments (Lines 23-24)
Line 156: why did you use only a fragment of the COI instead of the whole mitochondrial genome?
The partial cox1 sequences were used due to a lack of available data for the full mitochondrial genome for other aporocotylids. The methods have been updated to clarify this, as follows: “The partial cox1 region was chosen as full mitochondrial genomes were not available for most species.” (Lines 168-169)
Line 166; Indicate the total yield of reads from the sequencing run.
This has been added: “The final sequencing run for C. orientalis yielded a total of 5750829 reads, with a N50 of 4733 bp”. (Line 176-177)
Typically the WMG length in trematodes ranges from 14,000 to 16,000 base pairs. can you explain the unusual result?
We acknowledge that the mitochondrial genomes presented here are longer than what has previously been reported in trematodes. This is due to the non-coding region identified for both species, which was able to be identified due to the long-reads sequenced in this paper for C. orientalis, and previously for C. forsteri (as per Coff et al. 2022 – reference [13] in the manuscript). We believe we have sufficiently addressed this in the discussion of the manuscript, with the following statements:
“The length of the two mitogenomes is longer than what is reported for most trematodes, however this is mostly due to the extensive non-coding region identified, which is significantly larger than most previously reported NCR [45]. The coding region however, falls within the expected range, slightly longer than several species including Fasciola hepatica, Hypoderaeum conoideum, Echinostoma caprioni, Paragonimus westermani and Prosthogonimus cuneatus, but shorter than most schistosomes such as Schistosoma spindale, S. mansoni, S. japonicum and S. mekongi [45-47].”. (Lines 265-271)
“The identification of long non-coding regions is still only recent, with the development of long read sequencing allowing these regions to be elucidated without the need for assembly, which was difficult due to the repetitive nature of the NCR. The length of the C. orientalis NCR is within the range of long NCRs that have been identified so far: non-coding regions of 4–5 kb have been identified in S. bovis and Clonorchis sinensis, 6.9 kb in P. westermani, and 18.5 kb in S. haematobium [24,59,62]. The NCR of C. forsteri is now the longest NCR to be identified for a trematode to date at > 20 kb. Despite the variation between the two species, the identification of these NCR is highly confident as they were identified without the need for assembly. For both species, the whole NCR was confirmed through a consensus of reads mapping along the entire region. In C. orientalis several thousand reads were identified that covered the full length of the region, however even in C. forsteri in which no amplification was done, 5 reads covered the full length of the non-coding region.” (Lines 317-329)
Were assembly and circularization validated?
Yes, the assembly and circularization were validated for both species. For the purpose of easy comparison between species, the linear view of the mitochondrial genomes were presented in the main manuscript (Figure 1), however a figure of the circular mitochondrial genomes was provided in the supplementary material (Figure S1).
There is no information about the short vs long read results in the case of Cardicola forsteri.
Raw sequencing results were not reported for C. forsteri in this manuscript, as these results were previously covered in the paper by Coff et al. (2022).
Round 2
Reviewer 4 Report
Comments and Suggestions for Authors
Dear Editor,
The authors have made all the necessary corrections and have adequately addressed all comments in the revised version of the manuscript. I have no further comments and believe that the manuscript is now suitable for publication.
Best regards